# Artificial Intelligence Assisted Computational Tomographic Detection of Lung Nodules for Prognostic Cancer Examination: A Large-Scale Clinical Trial

**DOI:** 10.3390/biomedicines11010147

**Published:** 2023-01-06

**Authors:** Heng-Sheng Chao, Chiao-Yun Tsai, Chung-Wei Chou, Tsu-Hui Shiao, Hsu-Chih Huang, Kun-Chieh Chen, Hao-Hung Tsai, Chin-Yu Lin, Yuh-Min Chen

**Affiliations:** 1Department of Chest Medicine, Taipei Veterans General Hospital, Taipei 112, Taiwan; 2Faculty of Medicine, School of Medicine, National Yang Ming Chiao Tung University, Taipei 112, Taiwan; 3Division of Thoracic Surgery, Department of Surgery, Chung Shan Medical University Hospital, Taichung 40201, Taiwan; 4Institute of Medicine, College of Medicine, Chung Shan Medical University, Taichung 40201, Taiwan; 5Division of Pulmonary Medicine, Department of Internal Medicine, Chung Shan Medical University Hospital, Taichung 40201, Taiwan; 6Department of Applied Chemistry, National Chi Nan University, Nantou 545301, Taiwan; 7Department of Medical Imaging, Chung Shan Medical University Hospital, Taichung 40201, Taiwan; 8School of Medicine, College of Medicine, Chung Shan Medical University, Taichung 40201, Taiwan; 9Institute of New Drug Development, College of Medicine, China Medical University, Taichung 40402, Taiwan; 10Tsuzuki Institute for Traditional Medicine, College of Pharmacy, China Medical University, Taichung 40402, Taiwan; 11Department for Biomedical Engineering, Collage of Biomedical Engineering, China Medical University, Taichung 40402, Taiwan

**Keywords:** lung nodule, computed tomography, artificial intelligence, computer-assisted detection

## Abstract

Low-dose computed tomography (LDCT) has emerged as a standard method for detecting early-stage lung cancer. However, the tedious computer tomography (CT) slide reading, patient-by-patient check, and lack of standard criteria to determine the vague but possible nodule leads to variable outcomes of CT slide interpretation. To determine the artificial intelligence (AI)-assisted CT examination, AI algorithm-assisted CT screening was embedded in the hospital picture archiving and communication system, and a 200 person-scaled clinical trial was conducted at two medical centers. With AI algorithm-assisted CT screening, the sensitivity of detecting nodules sized 4–5 mm, 6~10 mm, 11~20 mm, and >20 mm increased by 41%, 11.2%, 10.3%, and 18.7%, respectively. Remarkably, the overall sensitivity of detecting varied nodules increased by 20.7% from 67.7% to 88.4%. Furthermore, the sensitivity increased by 18.5% from 72.5% to 91% for detecting ground glass nodules (GGN), which is challenging for radiologists and physicians. The free-response operating characteristic (FROC) AI score was ≥0.4, and the AI algorithm standalone CT screening sensitivity reached >95% with an area under the localization receiver operating characteristic curve (LROC-AUC) of >0.88. Our study demonstrates that AI algorithm-embedded CT screening significantly ameliorates tedious LDCT practices for doctors.

## 1. Introduction

Health examination using CT or LDCT has emerged as a popular method for the early detection of lung cancer [1]. However, very small or early lung cancers that cannot be easily detected by regular chest radiography are more likely to be examined using LDCT [2,3]. With the prevalence of CT or LDCT implementation in health examinations, the mortality rate resulting from lung cancer and any cause decreased by approximately 20% and 7%, respectively, in patients who underwent chest CT or LDCT examination compared with those who underwent only single-view posteroanterior chest radiography [4]. Furthermore, the CT or LDCT practices were approved by Medicare for lung cancer screening for smokers in 2015 in the United States [5]. Thus, the superiority of CT for the early detection of lung cancer has been demonstrated. Consequently, many academic and radiological communities have recently added LDCT practices to regular screening services for lung cancer detection. Their evidence suggests that LDCT screening for lung cancer detection results in a favorable balance between the advantages and disadvantages [6]. In March 2021, the United States Preventive Services Task Force (USPSTF) published the revised guidelines that recommends annual LDCT screening for lung cancer detection in adults aged 50–80 years with a 20-pack-year smoking history or more and those who currently smoke or have quit within the past 15 years, highlighting the standardization of this imaging examination methodology in clinical scenarios [7].

Principally, this CT or LDCT image examination scenario detects lung nodules, which can become malignant depending on the nodule type and size [1,8,9]. However, radiologists who evaluate a large number of slides in one LDCT screening face tremendous challenges, such as mechanical repetition, detailed examination, and tedious work leading to the easy omission of small nodules [10] and a lack of consistent criteria [11,12]. Therefore, several modified imaging check methodologies have been developed to alleviate the misjudgment of small lung nodules [13,14], such as computer-aided detection (CADe) and artificial intelligence (AI) algorithms, to ease the search for lung nodules [15,16,17,18,19,20,21,22,23,24,25,26]. 

Medical imaging is a private domain that is not mutually communicated between hospitals. Each hospital has independent clinical training protocols for doctors. Therefore, it is essential to explore the compatibility and performance of CT image detection assistance-oriented AI software in different medical centers or areas. In this study, we first collaborated with a group, V5 Technologies, which addressed the problems encountered in electronic industrial production through innovative auto-optical inspection techniques. They developed an AI algorithm for the autodetection of lung nodules from CT images, V5-MED-LU01, which was installed in the central computer system of the hospital and connected to the original picture archiving and communication system (PACS). Doctors could read the results on the original user interface or their favored Digital Imaging and Communications in Medicine (DICOM) viewer.

This study aimed to determine whether AI algorithm support improves the diagnostic performance and accuracy in identifying and marking nodules on chest CT images. The diagnostic performance improvement was measured throughout the study in terms of the area under the receiver operating characteristic (ROC) curve (AUC), sensitivity, and specificity among the doctors assisted by V5-MED-LU01 for detecting pulmonary nodules. We hope to make the best effort to regularly examine and monitor lung nodule deformation and gradually improve the detection rate of early-stage lung cancer. 

## 2. Materials and Methods

### 2.1. Clinical Trial Design of AI-Assisted Chest CT Examination

This was a retrospective, multiple-reader, multiple-case (MRMC) investigation conducted between 2018 and 2020 at two medical centers, Taipei Veterans General Hospital (TVGH) and Chung Shan Medical University Hospital (CSH), in Taiwan. We utilized a comparator reader study involving assisted reading of chest CT images using the AI algorithm detection software, V5 Pulmonary Image Computer-Aided Detection Software, V5-MED-LU01 (Version:1.0.1, V5 Technologies Co., Ltd., Hsinchu City, Taiwan), and unassisted reading of chest CT images that met the image quality requirements. Historical chest CT/LDCT imaging records were collected from 200 patients, 100 patients from each medical center, with approximately half of the pathologically confirmed to be pulmonary nodules and half confirmed to be typical CT images, as described in the Results section.

Four doctors from each medical center participated as clinical readers in the study. An expert panel of three senior, board-certified specialists (including pulmonologists, chest surgeons, and radiologists) with fifteen years or more of practice were entrusted as the clinical truthers to establish the “Ground Truth”. After collecting the patient CT images as per the eligibility criteria, the clinical truthers first established the ground truth of the CT image of each patient by determining whether pulmonary nodules were present on the CT images. The differences in the clinical interpretation among the three experts were resolved based on the majority rule (i.e., the same clinical judgment from two of the three specialists was ruled as the Ground Truth). All experimental protocols are illustrated (Figure 1).

After the Ground Truth was established for the CT image of each subject, the samples were randomly sorted and provided to the clinical readers for identification and marking. All four junior doctors were trained to achieve maximum consistency in interpretation before trial initiation.

This MRMC was a fully crossed design study where all doctors independently read all cases under multiple reading modalities (i.e., readers unassisted vs. readers assisted by the AI algorithm detection software). Each reader’s interpretation of the presence of the nodule, nodule location, nodule size, level of suspicion, and other information was recorded.

According to the aforementioned design, the reading process was divided into two reading sessions, with the test samples evenly divided into group A and group B. During the first reading session, the readers in group A were unaided by the AI algorithm detection software, whereas those in group B were aided by the AI software. After a wash-out period of at least four weeks, the two groups of images were read again, but the use of the AI software was reversed, that is, group A was assisted by the AI software, and group B was unassisted by the AI software.

### 2.2. Inclusion Criteria for Clinical Trial Design

The medical situation and chest CT images of patients with the following characteristics were considered eligible for inclusion in this study: patients over 20 years of age of either sex with chest CT/LDCT images showing ≥1 pulmonary nodule or no nodule, with the nodule size ranging from 4 to 32 mm (inclusive range); the CT/LDCT images containing the entire lungs bilaterally in DICOM format; and a sample slice thickness of <3.0 mm (non-recombinant image), >100 continuous slices without gaps, and in-plane resolution of 512 × 512 pixels. The medical imaging equipment used in this study was based on data available from the two medical centers.

### 2.3. Exclusion Criteria for Clinical Trial Design

Patients with the following medical conditions were excluded from the study: chest CT/LDCT images showing apparent abnormalities, such as atelectasis, pneumothorax, significant pulmonary infiltrates, prominent pulmonary fibrosis, pneumonia, and interstitial lung disease; patients with lung tumors and associated abnormalities, such as a significant lung mass (>32 mm), diffuse lung metastasis, massive pleural effusion, or anticancer treatment-associated pneumonitis; and patients with concurrent cancers, other than lung cancer. 

### 2.4. AI algorithm Embedded CT System and Hospital Connection Structure

The AI algorithm composed of convolutional neural networks used in this study complies with the standard DICOM protocol for receiving, processing, and transmitting chest CT images. The AI system was installed on the server host and connected to the hospital PACS via the intranet, as illustrated (Figure 2). Once the AI system receives a series of CT images from the PACS system, it evaluates whether the image quality meets the requirements and confirms whether the required fields are available. It then identifies and marks the regions of interest (ROI) for the suspected pulmonary nodules. After the marking is complete, the information, in conformance with the standard DICOM protocol, is transferred back. Medical specialists then used their DICOM viewer to read the image with the detection results displaying the marked suspected nodules.

### 2.5. Evaluation and Record

The performance of the doctors in detecting the lung nodules was evaluated based on the nodule-level area under curve (AUC) [3]. The doctors were instructed to mark and rate all suspicious nodules on a 100-point level of suspicion (LOS). A higher LOS rating represents greater confidence of the reader in indicating a nodule, providing a reference for medical follow-up. The nodule contour was drawn as an ROI to annotate the site and size. Nodule localization requires the center of the doctor’s mark of a suspicious nodule to fall within the radius of an actual nodule based on the ground truth. 

This study collected the AUC, sensitivity, and specificity to collectively determine the performance of the CT image prediction scenario with or without AI assistance. The comparison between the AUC with and without the AI assistance was performed using the analysis of variance (ANOVA) test with alpha = 0.05. The analysis of the variance method proposed by Obuchowski and Rockette [27] and later enhanced by Hillis [28] was used to estimate the ROC curves, AUC, and 95% confidence interval (CI), after accounting for localization. In addition, one or more nodules marked in the same subject by a reader were analyzed as independent data. The superiority of AUC was demonstrated if the upper limit of the two-sided 95% CI for the difference in AUC without AI assistance between AUC with AI assistance was below zero. The AUC, sensitivity, and specificity were calculated using the following equation in accordance with a previous publication [29]:$$AUC=Integrated\;the\;trapezoidal\;area\;under\;the\;localization\;ROC\;\left(LROC\right)\;curve$$
$$Sensitivity=\frac{True\;Positive}{True\;Positive+False\;Negative}$$
$$Specificity=\frac{True\;Negative}{True\;Negative+False\;Positive}$$

For the localization ROC (LROC) curve, ROC analyses were conducted via logistic regression stratified by the reader and modality, with nodule-level LOS as an independent variable and the ground truth of each nodule as a dependent variable, to obtain the parameter estimates and ROC output. The ROC output includes the probability of a nodule at each specified LOS and each pair of sensitivity and 1-specificity. 

### 2.6. Statistical Analysis

Data are presented as mean ± standard deviations (SDs), and statistical comparisons were performed using Student’s *t*-test or one-way ANOVA. *p* values < 0.05 were considered statistically significant and labeled as *, whereas *p* values < 0.01 and <0.001 were labeled as ** and ***, respectively. All calculations were performed using the Statistical Analysis System (SAS) licensed to the China Medical University. 

## 3. Results

### 3.1. AI-Embedded CT Screening Marks the Varied Nodules in the Regular LDCT Images Efficiently

To investigate whether AI-assisted CT examination can efficiently mark the nodules embedded in the CT slice images prior to formal examination by the doctor, the bilateral chest CT images with varying nodule sizes, sex, and age were collected. Chest CT imaging records of 200 patients were collected from TVGH and CSH. Each medical center included 100 patient examinations. The principal characteristics, including age, sex, and the number of patients with detected pulmonary nodules from each medical center, CSH (Appendix A) and TVGH (Appendix A), are summarized. At each medical center, we included 49 patients with at least one detected nodule and 51 with currently normal CT images without detected nodules. At CSH and TVGH, patients aged 51–70 years were the majority, accounting for 82% and 70% (Appendix A), respectively. 

In CSH, 49 of the 100 examined patients had at least one or more pulmonary nodules on their CT images. A total of 107 nodules were detected, with sizes ranging from 4 mm to ≥16 mm, with the majority being between 5 mm and <6 mm (30 out of 107) (Appendix A). Similarly, in TVGH, at least one or more pulmonary nodules were detected during CT examination in 49 patients. A total of 98 nodules with sizes ranging from 4 mm to ≥29 mm, with the majority being between 9 and <10 mm (15 out of 98) (Appendix A) were detected. In addition, the current study was conducted at two medical centers with multi-CT machines and varying slice thicknesses, as shown in the Appendix A), which revealed an obvious difference. Remarkably, the thinner slices used in TVGH occupied 71% (Appendix A), resulting in a tremendous quantity of DICOM slices in each CT examination, leading to a heavy burden on the medical image reading.

To demonstrate that AI-assisted CT screening could be applied to multi-type lung nodule recognition and can precisely mark the hazy and suspicious nodule in advance, we mainly selected several paired CT slices that had a discrepancy between AI engagement and doctor-alone examination for comparison. The representative CT images showed that AI-assisted CT screening could detect various lung nodules, including solid, part-solid, and ground glass nodules (GGN), which are the most ambiguous nodule types that trouble pulmonary radiologists (Figure 3). In the two left columns, we compared identical nodules that were not detected by doctors but were successfully detected with AI assistance. The two right columns show identical nodules recognized by the doctors and AI while annotating the probability and nodule size (Figure 3). These data demonstrate the superiority of AI assistance, which can efficiently find and annotate the lung nodule from several CT slices, saving the doctors’ interpretation time.

### 3.2. Sensitivity of Nodule Detection Ameliorated by AI-Assisted CT Screening

All nodules of 98 patients from two medical centers were classified into four categories (Table 1). A category for nodule sizes < 5 mm was included to determine the performance of AI-assisted CT examination in detecting nodules of different sizes. The detection sensitivity with or without AI assistance was calculated (Table 1). The data revealed that the nodule size ranging from 6 to 10 mm occupied the majority. The overall nodule detection sensitivity in the scenario of AI standalone reached 95.6% (Table 1). The detection sensitivities of the categories containing pulmonary nodules < 20 mm were increased significantly with AI assistance. Notably, nodules sized between 4 and 5 mm showed increased detection sensitivity with tremendous significance. The results demonstrate the superiority of AI-assistance CT screening in detecting tiny nodules, which is conventionally troublesome for pulmonologists. 

The nodules were further classified into three categories according to their radiological appearance. GGN, which is conventionally troublesome to recognize through CT images, occupied the majority and accounted for 65% of the total nodules. Our data revealed that the sensitivity of detecting the three types of nodules, including solid, part-solid, and GGN, was significantly ameliorated in both medical centers with distinct CT machines and pulmonary professional interned training processes (data not shown). Notably, the sensitivity of detecting GGN increased by 18.5% to achieve a sensitivity of 91% in the AI-assisted CT screening. Furthermore, the sensitivity of detecting the solid-type nodules was vastly elevated; more doctors could detect the nodules with AI assistance (Appendix A) as the PACS-embedded AI algorithm can simultaneously fine-tune the grey contrast of DICOM for efficient recognition. 

### 3.3. AI Assistance Significantly Augmented the Sensitivity of Nodule Detection but Maintained the Specificity for Regular LDCT

Since AI assistance tremendously ameliorates the sensitivity of detecting small nodules in CT images and provides a reference for doctors’ interpretation, the AI detection accuracy, such as false positive and false negative ratios, was addressed. The four junior doctors’ examination results were compared with the ground truth data collected from the medical centers and statistically analyzed according to the equation shown in the Materials and Methods section. Sensitivity reveals the ratio of doctors misjudging the “false negative,” reflecting whether the doctor underestimates the “true positive” nodule on the CT images. All eight junior doctors from both medical centers had a higher nodule-detecting sensitivity with the assistance of the AI algorithm. The average sensitivity significantly increased from 67.7% without AI assistance to 88.4% with AI assistance (*p* < 0.001) (Figure 4a). Specificity reveals the ratio of doctors misjudging the “false positive,” reflecting the accuracy of predicting the “true negative” on the CT images. The data revealed that the specificity was slightly lower for detection with AI assistance than that without it; however, no statistically significant difference (*p* > 0.05) was observed. The average specificity decreased from 89% without AI assistance to 87% with AI assistance. Our data demonstrated that AI assistance significantly augmented the sensitivity of nodule detection without sacrificing specificity, thereby maintaining a high recognition ratio for “true negatives”.

### 3.4. Clinical Performance and Effectiveness of Nodule Detection in LDCT Examination Increased Significantly with AI Assistance

To evaluate the performance and advancement of AI assistance in predicting the nodules accurately, the AUC values of the doctors alone and doctors assisted by AI were compared. The AUCs were based on the level of suspicion (LOS) scores of the nodules requiring correct localization, where the central point of each junior doctor’s mark of a suspicious nodule fell within the radius of an actual nodule based on the ground truth. The AUCs with and without the pre-engagement of the AI algorithm in CT image recognition for each doctor are shown in Figure 5. The data showed that all eight doctors had greater AUCs when using AI algorithms. The average AUC significantly increased from 0.684, without using the AI algorithm, to 0.883 with the use of the AI algorithm (*p* < 0.001). The average difference in AUC was approximately 0.2, demonstrating the superiority of the AUC for doctors assisted by AI algorithms over those without.

Furthermore, to determine the AI score for doctors to use in the AI algorithm engaged in CT image recognition, the CT image stacks of the patients were recognized by the AI algorithm alone, and a free-response operating characteristic (FROC) curve was generated. The FROC curve is a plot of the sensitivity versus the false-positive (FP) rate per patient, which is a benchmark of the AI-assisted detection system performance and accuracy. Another indicator that reflects the accuracy and performance of lung nodule detection is the localization receiver operating characteristic (LROC) curve. All participating doctors were instructed to mark and rate all suspicious nodules, with or without AI assistance. Nodule localization requires the center of a reader’s mark of a suspicious nodule to fall within the radius of an actual nodule based on the ground truth. The frequency distribution of these ratings was used to construct two LROC curves: one for the interpretations made without the engagement of the AI algorithm and one for those made with the engagement of the AI algorithm.

Data were collected from both medical centers for analysis of the FROC and LROC plots, revealing the FROC curve when the AI score was ≥0.4, the FP rate per CT study was 0.3, and the sensitivity reached 95.6% (Figure 6a). The LROC trapezoid area in AI-assisted CT screening was significantly superior to that of the area without AI assistance (Figure 6b). Therefore, based on the FROC curve, the AI score was set as 0.4 for doctors’ examinations with AI assistance throughout this study. 

## 4. Discussion

As previously described, the healthcare data and medical images from distinct medical centers, even those with an identical hospital system but distinct branches, are not allowed to exchange freely. Furthermore, the training progress of the attending physicians or radiologists in distinct medical centers may be different, resulting in the human perception bias of lung nodule recognition emerging as a critical issue in lung cancer screening [30,31,32]. Consequently, to address the insufficient accuracy of lung nodule detection from traditional CT images, advanced three-dimensional (3D) display methods [30,32], novel blood specimen examinations [33] and AI-assisted image prediction [15,16,17,18,19,20,21,22,23,24,25,26] have been developed to assist pulmonologists. Therefore, the compatibility of AI systems with different hospitals and well-trained pulmonary professionals is a critical issue. Our data demonstrated that the V5-MED-LU01 AI software is compatible with the distinct PACS of the two medical centers (Figure 1 and Figure 2) and multi-type CT scanning machines with various resolutions and slice thicknesses (Appendix A). Furthermore, the detection rates of distinct sizes and types of lung nodules were comprehensively improved with AI assistance (Table 1 and Figure 3). The detection rate of nodules smaller than 5 mm was increased significantly, which is conventionally considered difficult for AI prediction [3]. The nodule detection sensitivity of GGN also increased significantly. GGN is the most dangerous type of lung nodule in East Asia and is prone to tumorigenesis depending on its size [1]. 

In contrast, although our data demonstrate that AI assistance significantly augmented the sensitivity of nodule detection (Figure 4a), the specificity was not sacrificed and still maintained a high recognition ratio for “true negative” (Figure 4b). In a previous study, AI assistance ameliorated the specificity but reduced the sensitivity of nodule recognition [29]. Their study developed a deep learning (DL) algorithm to identify pulmonary nodules that appeared on LDCT images. Their data supported the DL model and revealed superior performance compared with the average performance of the radiologists. The AUC value of the DL model exceeded the average performance of the radiologists. However, the sensitivity of the DL model was lower than that of the radiologists, reaching 10% [29].

The AUC value revealed the superiority of AI assistance in nodule detection over doctors alone in both medical centers in this study, reaching a value of approximately 0.88, which is superior to that reported in a previous publication [3]. Lo et al. utilized a United States Food and Drug Administration (FDA) certified software system (ClearRead CT Vessel Suppression, Riverain Technologies, Miamisburg, OH, USA) to detect small pulmonary nodules from chest CT image stacks collected from 324 cases in clinical trials. Their data revealed that the sensitivity for detecting nodules no smaller than 5 mm was approximately 82% in the standalone test. The LROC-AUC value for detecting clinically actionable nodules increased from 0.584 when unaided by the AI algorithm to 0.692 when aided by the AI algorithm [3]. However, their data showed that the sensitivity of the standalone test and the value of LROC-AUC were still lower than those observed in the current study, reaching 95.6% (Table 1) and 0.883 (Figure 5), respectively. In addition, their specificity decreased from 89.9% to 84.4%. Conversely, our data showed an increase in the sensitivity but no compromise in the specificity (Figure 4). Furthermore, our data show superiority in detecting nodules smaller than 5 mm or GGN type with highly irregular and ambiguous opacity, which remains a diagnostic challenge [9]. 

The FP rate indicates the misidentification of nodules, either by a human or by machine, and may lead to inappropriate subsequent medical action [34] and consideration of further malignancy. The factors that result in FP identification include the focal area of pneumonia, granuloma from prior infection, calcified granuloma, focal areas of lung or pleural scarring, and intrapulmonary lymph nodes [3]. Moreover, the interpretation of AI engagement is also critically affected by the CT image stack input quality, which needs a more sophisticated algorithm to pre-treat the CT images [35]. Since the FP rate can lead to the underestimation of GGN in CT screening, ameliorating the sensitivity and maintaining the specificity of detecting GGN has emerged as a critical medical problem. Nevertheless, data from large scale trails have demonstrated that lung cancer mortality was significantly lessened by the more prevalent LDCT screening [4]; the management of subsequent treatment from a doctor and hospital care should be more seriously considered [36,37]. Collectively, our data fulfill this unmet need.

Our data demonstrate that the V5-MED-LU01 AI software is compatible with various CT machines and PACS in different hospitals while assisting pulmonologists and significantly ameliorating lung nodule prediction. The AI algorithm can assist doctors in identifying and marking pulmonary nodules in chest CT image stacks and display the marking results for doctors to determine subsequent medical treatment and follow-up during a routine examination [38]. For example, in patients with previously treated lung cancer, a newly detected nodule most likely represents distant metastasis from the initial or second primary lung cancer. The improved sensitivity of CT in detecting nodules should guide decisions regarding biopsy, composition, and subsequent medical treatments. The AI algorithm used in CT examination is not intended to be used as the sole basis for diagnosis. Therefore, it is not possible to simplify, replace, or substitute for, in whole or in part, the healthcare provider’s judgment and diagnostic analysis or procedure in the future. Hopefully, the AI-assisted CT system will make the best effort to regularly examine and monitor lung nodule deformation and gradually improve the detection rate of early-stage lung cancer.

Currently, we use the V5-MED-LU01 service in our outpatient clinic for unreported CT or LDCT studies of the chest, particularly for CT images that patients bring in from other hospitals. It helps to quickly annotate pulmonary nodules and saves time for busy physicians. Physicians are very satisfied with its performance as an immediate aid in the outpatient clinic and as a backup. Additionally, we are conducting a longitudinal study to evaluate changes in lung tumors using the V5-MED-LU01′s annotation and measurement functions as the core of the study. In order to improve the software in the future, we are working on a function that allows us to match images of the same node from different examination dates, which will enhance our ability to track and compare lung tumors. Our team is working on expanding the capabilities of the V5-MED-LU01 in order to make it more useful for clinicians. The developing functions, including differentiating between benign and malignant nodes, detecting lymph nodes, and identifying other lesions that physicians may easily miss, can improve the accuracy and efficiency of diagnosis and treatment. These features may be especially useful in busy outpatient clinics and have the potential to make a positive impact on patient care.

## 5. Conclusions

The objective of this study was to evaluate the doctors’ performance in identifying pulmonary nodules on chest CT when assisted by AI software, V5-MED-LU01. The results of this study showed that the overall increase in the AUC values was statistically significant when assisted with AI software, despite the use of various DICOM files of CT scanners and nodule sizes and types, including solid, GGN, and part-solid GGN. The use of AI software for pulmonary nodule detection is efficient and valuable for lung cancer screening.

## Figures and Tables

**Figure 1 biomedicines-11-00147-f001:**
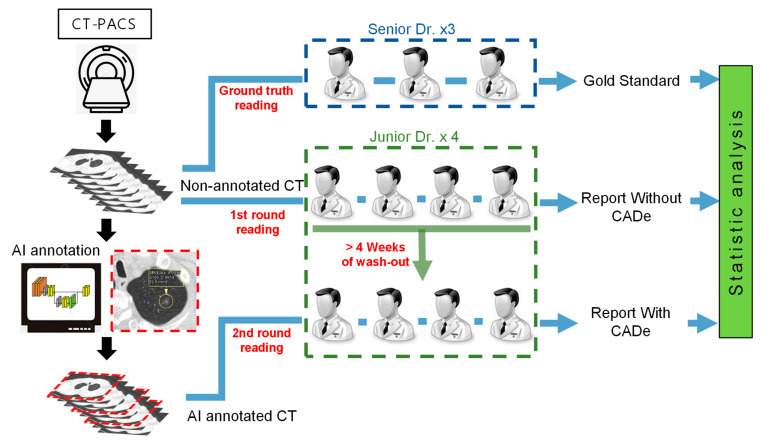
Scheme of the clinical trial of chest CT image examination assisted by AI algorithm in each medical center. The left part demonstrates the simple schema of the image processing flow that produces the non-annotated and annotated CT. Part B demonstrates the multiple-reader multiple-case study, which included four junior doctors in each medical center as clinical readers. In addition, an expert panel of three senior board-certified senior doctors served as the clinical truthers to establish the Ground Truth. The reading process was divided into two reading sessions. During the first reading session, the readers were unaided by AI algorithm detection software. After a wash-out period of at least four weeks, the images were read again with the aid of V5 Pulmonary Image Computer-Aided Detection Software, V5-MED-LU01 (Version:1.0.1, V5 Technologies Co., Ltd., Hsinchu City, Taiwan).

**Figure 2 biomedicines-11-00147-f002:**
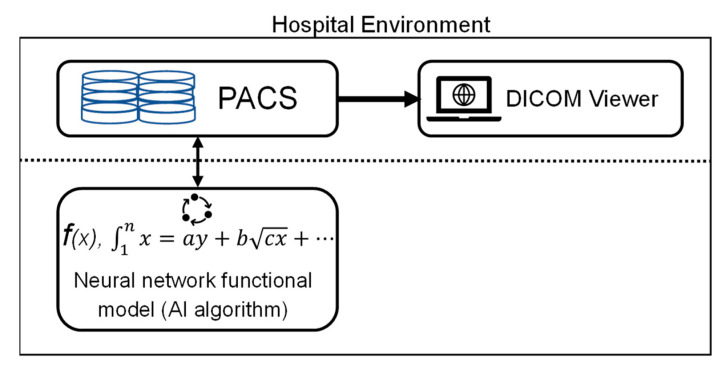
AI algorithm embedded in the PACS infrastructure of the hospital. AI: artificial intelligence; DICOM, digital imaging and communications in medicine; PACS, picture archiving and communication system.

**Figure 3 biomedicines-11-00147-f003:**
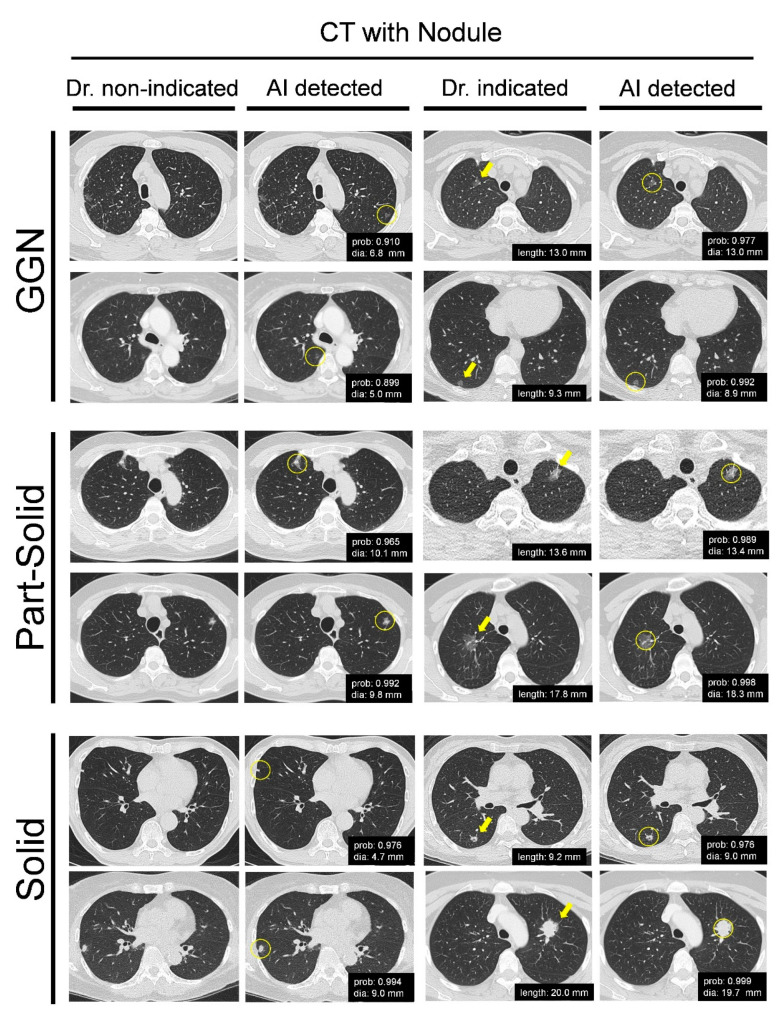
Comparison of the lung nodules detected by doctors with or without AI assistance. Multi-type lung nodules, including GGN, part-solid, and solid, were detected by the doctors with or without AI assistance in advance, and the paired CT images with discrepancies between AI engagement and doctor-alone examination were selected for comparison. The two left columns show that the slices annotated by AI software but not indicated by doctors. The two right columns show the slices simultaneously annotated by AI software and indicated by doctors. The representative images show the paired CT images in the same slice number. The yellow circles indicate pulmonary nodules detected by AI. The yellow arrows indicate pulmonary nodules found by physicians. AI, artificial intelligence; CT, computed tomography; GGN, ground glass nodules.

**Figure 4 biomedicines-11-00147-f004:**
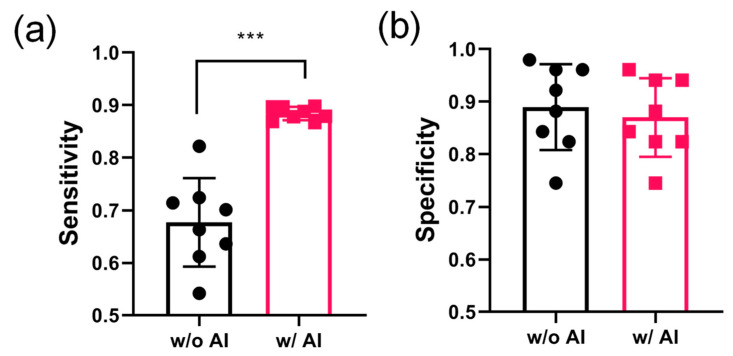
The sensitivity and specificity of lung nodule detection ameliorated by AI-assisted CT screening. The eight junior doctor’s examination results compared with the “Ground Truth” data collected from both medical centers were statistically analyzed. (**a**) Sensitivity; (**b**) specificity. Data represents mean ± SD. *** *p* < 0.001. AI, artificial intelligence; CT, computed tomography; SD, standard deviation.

**Figure 5 biomedicines-11-00147-f005:**
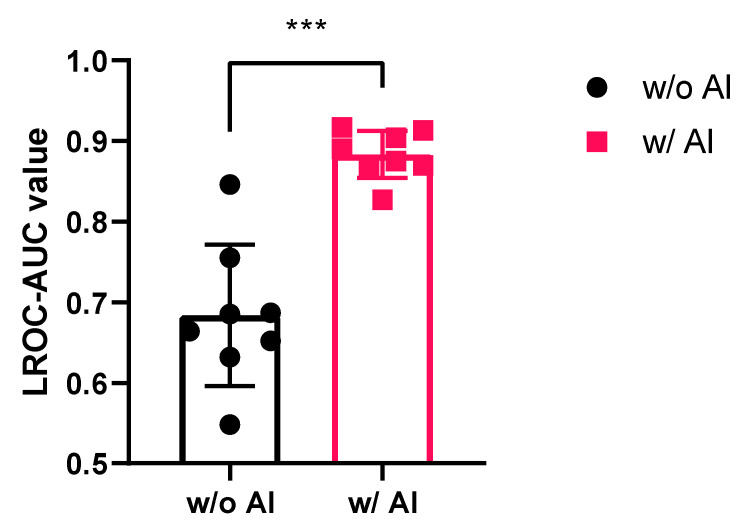
LROC-AUC value increased for AI-assisted CT screening. LROC-AUC value is calculated according to the LOS scores of the nodule, where the central point of each junior doctor’s mark of a suspicious nodule falls within the radius of an actual nodule based on the ground truth. The AUC values with and without the pre-engagement of AI algorithm in CT images’ recognition in CSH and TVGH were statistically analyzed. Data represent the mean ± SD. *** *p* < 0.001. LROC, localization receiver operating characteristic; AUC, receiver operating characteristic curve; LOS, level of suspicion; AI, artificial intelligence; CT, computed tomography; TVGH, Taipei Veterans General Hospital; CSH, Chung Shan Medical University Hospital.

**Figure 6 biomedicines-11-00147-f006:**
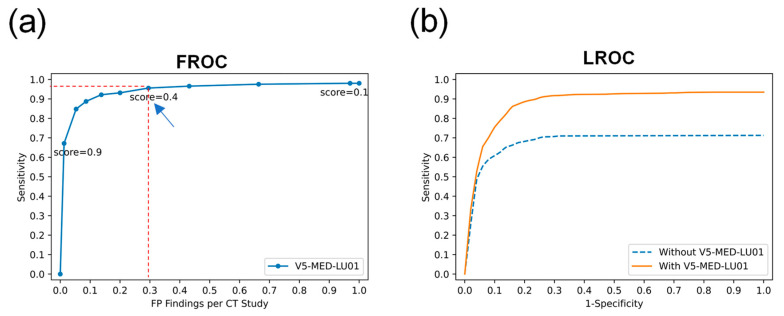
FROC and LROC curves to determine the superior sensitivity of AI-assisted CT screening. (**a**) The AI algorithm alone recognized the CT image stack of the patient, and the FROC curve was generated. (**b**) To generate the LROC curve, all doctors were instructed to mark and rate all suspicious nodules with or without AI assistance. The arrow points to the point where the AI score was greater than or equal to 0.4, while the false positive rate per CT study was 0.3, and the sensitivity reached 95.6%. FROC, free response operating characteristics; LROC, localization receiver operating characteristic; CT, computed tomography.

**Table 1 biomedicines-11-00147-t001:** Summary of the nodule sizes detected with or without AI assistance.

Nodule Size (mm)	Number of Nodules	Sensitivity			
		Standalone	Reader Study		
			w/o AI	w/AI	*p*-Value
4~5	69	92.8%	39.3%	80.3%	0.0003
6~10	99	96.0%	80.8%	92.0%	0.002
11~20	29	96.6%	83.1%	93.4%	0.042
>20	8	100.0%	71.9%	90.6%	0.221
Total	205	95.6%	67.7%	88.4%	0.0002

Data were collected and mixed from two medical centers participating in this study.

## Data Availability

All data are disclosed in the manuscript and Appendix A.

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
