# Peer review of "Artificial Intelligence Assisted Computational Tomographic Detection of Lung Nodules for Prognostic Cancer Examination: A Large-Scale Clinical Trial"

_biomedicines, 2023, doi:10.3390/biomedicines11010147_

Round 1
Reviewer 1 Report
I want to thank the handling editor for giving me the opportunity to review the manuscript entitled “Artificial intelligence assisted computational tomographic detection of lung nodules for prognostic cancer examination: a large-scale clinical trial” by Chao and colleagues. This is a multicentre, retrospective study investigating the use of artificial intelligence (AI) for assisting the detection of lung nodules on thoracic computed tomography (CT). The study included 200 participants with 205 lung nodules in total. The overall sensitivity of detecting nodules sized at least 4 mm and 4–5 mm increased by 21% and 41%, respectively, in the AI algorithm-assisted CT screening. Furthermore, the sensitivity increased to 88% overall and 91% for ground glass nodules. The free-response operating characteristic AI score was ≥0.4, and the AI algorithm standalone CT screening sensitivity reached >95%, with an area under the localization receiver operating characteristic curve of >0.88. Therefore, the authors conclude that AI algorithm-embedded CT screening significantly improved the detection of lung nodules. This is an interesting and informative study that has the potential to be a valuable addition to the relevant literature.
This is an overall well written paper. The introduction sets the appropriate background even for the reader with little knowledge on the topic. The methods are adequately described to allow replication of the results, which are clearly presented with relevant tables and figures. The findings are discussed within the context of the pertinent literature, and the conclusions are based on the results of the study. Here, I have made a few suggestions that (in my opinion) could help improve the overall quality of the manuscript.
· In the Abstract, the authors may consider presenting more information regarding the number of the patients included and the examined lung nodules (e.g., number, sizes, solid or subsolid). They may also consider reporting effect sizes for the outcomes of interest.
· The authors may consider reporting if the included patients were consecutive. If the participants were not consecutive, they may consider reporting how the patients were selected.
· The authors may consider reporting the indications for chest CT of the included participants (e.g., lung cancer screening programme, investigation of respiratory symptoms, surveillance imaging for cancer, etc.).
· The authors excluded patients with various other concurrent chest conditions. What is the reason for that? Do these pathologies impede the performance of the AI algorithm in detecting lung nodules?
· The authors may consider discussing how the implementation of the AI algorithm in clinical practice would have changed the management of the included patients.
· The authors may consider discussing limitations of their study.
Author Response
Point 1: In the Abstract, the authors may consider presenting more information regarding the number of the patients included and the examined lung nodules (e.g., number, sizes, solid or subsolid). They may also consider reporting effect sizes for the outcomes of interest.
Response: We appreciate the reviewer’s comment and valuable suggestion. We have accordingly revised the abstract and added more information, data, and exact experimental values. The revised Abstract was marked by word tracking as follows in line 34 of the manuscript:
“…centers. With AI algorithm-assisted CT screening, the sensitivity of detecting nodules sized 4–5 mm, 6~10 mm, 11~20 mm, and > 20 mm increased by 41%, 11.2%, 10.3%, and 18.7%. Remarkably, the overall sensitivity of detecting varied nodules increased by 20.7% from 67.7% to 88.4%. Furthermore, the sensitivity increased by 18.5% from 72.5% to 91% for detecting ground glass nodules (GGN), which is challenging for radiologists and physicians. The free-response operating characteristic (FROC) AI score was ≥ 0.4, and the AI algorithm standalone CT screening sensitivity reached > 95%...”
Point 2: The authors may consider reporting if the included patients were consecutive. If the participants were not consecutive, they may consider reporting how the patients were selected.
Response: We appreciate the reviewer’s comment and valuable suggestion. The CT scans of the chest were obtained from two hospitals that participated in this study. We randomly selected dates between 2018 and 2020 and reviewed all CT reports for pulmonary nodules on those days until reaching the target case numbers. If the CT study met the inclusion criteria and had no exclusion criteria, it was included in the study. Exclusion criteria included concurrent chest conditions that may obscure or cause phantom lung nodules. Since one AI cannot solve all types of pathologies, we focused on the task of detecting nodules in this study.
Point 3: The authors may consider reporting the indications for chest CT of the included participants (e.g., lung cancer screening programme, investigation of respiratory symptoms, surveillance imaging for cancer, etc.).
Response: Thanks for your comment and valuable suggestion. As the words in the point 2, we randomly selected CT scans from our CT worklist. The CT images included in our study are not come from cancer screening program or limited to single clinical purpose.
Point 4: The authors excluded patients with various other concurrent chest conditions. What is the reason for that? Do these pathologies impede the performance of the AI algorithm in detecting lung nodules?
Response: We appreciate the reviewer’s comment and valuable suggestion. And YES, many conditions may have great influence to the work of AI. In our experience, old granulomatous inflammation, bronchiectasis with secondary bacterial infection, and interstitial lung disease could cause much false positive readings. And large lobar pneumonia, lung collapse and pleural effusion may obscure small lung nodules. Since one AI cannot solve all types of pathologies, we focused on the task of detecting nodules in this study.
Point 5: The authors may consider discussing how the implementation of the AI algorithm in clinical practice would have changed the management of the included patients.
Response: We appreciate the reviewer’s comment and valuable suggestion. The software V5-MED-LU01 have been implemented in our outpatient clinic. We also have further effort to make it better. We make the feedback opinion from physicians and our further works clearer at the end of discussion. The revised Conclusion was marked by word tracking as follows in line 424 of the manuscript:
“Currently, we use the V5-MED-LU01 service in our outpatient clinic for unreported CT or LDCT studies of the chest, particularly for CT images that patients bring in from other hospitals. It helps to quickly annotate pulmonary nodules and saves time for busy physicians. Physicians are very satisfied with its performance as an immediate aid in the outpatient clinic and as a backup. Additionally, we are conducting a longitudinal study to evaluate changes in lung tumors using the V5-MED-LU01's annotation and measurement functions as the core of the study. In order to improve the software in the future, we are working on a function that allows us to match images of the same node from different examination dates, which will enhance our ability to track and compare lung tumors. Our team is working on expanding the capabilities of the V5-MED-LU01 in order to make it more useful for clinicians. The developing functions, including differentiating between benign and malignant nodes, detecting lymph nodes, and identifying other lesions that physicians may easily miss, can improve the accuracy and efficiency of diagnosis and treatment. These features may be especially useful in busy outpatient clinics and have the potential to have a positive impact on patient care”
Reviewer 2 Report
This paper proposes and evaluates an approach for AI algorithm-assisted CT screening significantly increasing the detection rate. The paper is interesting and well presented. The evaluation methodology is appropriate, and the experiments are well designed and executed. The improvements are significant and interesting for the community.
Comments to improve the paper:
· The weakest part of the paper is the AI algorithm. The paper does not describe any aspect of this algorithm. I think, it is necessary to describe in detail the algorithm to understand the evaluation results.
· I think there several initials and abbreviations. I’d suggest including a glossary.
· In the discussion and conclusions, I’d appreciate some evaluations from the doctors regarding the inclusion of this AI algorithm.
Author Response
Point 1: The weakest part of the paper is the AI algorithm. The paper does not describe any aspect of this algorithm. I think, it is necessary to describe in detail the algorithm to understand the evaluation results.
Response: We appreciate the reviewer’s comment and valuable suggestion. As we know, V5-MED-LU01 is an AI-based lung nodule detector that uses DICOM image series from CT scanners as input and produces the coordinates and feature attributes of nodules as output. The AI model belongs to the CNN (Convolutional Neural Network) family, and utilizes the architecture of ResNet and UNet. ResNet addresses the vanishing gradient problem by using a reference to the previous layer to compute the output at a given layer. The output from the previous layer, called the residual, is added to the output of the current layer. By configuring different numbers of channels and residual blocks in the module, different ResNet models can be created, such as the deeper 152-layer ResNet-152. Since the input data is a 3D lung image, our ResNet is a 3D version. The U-Net architecture helps to recover feature maps to a higher resolution and has good detection performance. V5-MED-LU01 is a commercially available software package that is protected by patents and intellectual property. As a clinical trial hospital, we are not able to provide further details beyond those provided by the manufacturer. We apologize for any inconvenience this may cause.
Point 2: I think there several initials and abbreviations. I’d suggest including a glossary.
Response: We thank the reviewer’s generous comment and thorough check. We have carefully checked all abbreviations; the full names were noted before they first appeared in the manuscript. The edited abbreviations and full names were marked by word tracking.
Point 3: In the discussion and conclusions, I’d appreciate some evaluations from the doctors regarding the inclusion of this AI algorithm.
Response: We appreciate the reviewer’s comment and valuable suggestion. The software V5-MED-LU01 have been implemented in our outpatient clinic. We also have further effort to make it better. We make the feedback opinion from physicians and our further works clearer at the end of discussion. The revised Conclusion was marked by word tracking as follows in line 424 of the manuscript:
“Currently, we use the V5-MED-LU01 service in our outpatient clinic for unreported CT or LDCT studies of the chest, particularly for CT images that patients bring in from other hospitals. It helps to quickly annotate pulmonary nodules and saves time for busy physicians. Physicians are very satisfied with its performance as an immediate aid in the outpatient clinic and as a backup. Additionally, we are conducting a longitudinal study to evaluate changes in lung tumors using the V5-MED-LU01's annotation and measurement functions as the core of the study. In order to improve the software in the future, we are working on a function that allows us to match images of the same node from different examination dates, which will enhance our ability to track and compare lung tumors. Our team is working on expanding the capabilities of the V5-MED-LU01 in order to make it more useful for clinicians. The developing functions, including differentiating between benign and malignant nodes, detecting lymph nodes, and identifying other lesions that physicians may easily miss, can improve the accuracy and efficiency of diagnosis and treatment. These features may be especially useful in busy outpatient clinics and have the potential to have a positive impact on patient care”
Reviewer 3 Report
Good paper, my specific comments are as follows;
-In order to robustly characterise the performance of your model, you need to quote the Acc, Spec, Sens and AUC
-Ensure all acronyms in the paper are defined
-As part of your background section, provide a breakdown of the concept of imaging, the pathology in question as well as prior AI methods
-The explicit gap in literature and associated contributions need to be listed
-The proposed AI algorithm needs considerable expansion and description, in order to boost understanding and repeatability
-The resolution on Fig 3 needs to be improved
-Are there related studies to benchmark your current study with?
-It may be worth including a flow diagram for your proposed method
-the conclusion requires considerable bolstering with explicit points for further work
-Refs need considerable bolstering
Author Response
Point 1: In order to robustly characterise the performance of your model, you need to quote the Acc, Spec, Sens and AUC
Response: We appreciate the reviewer’s comment and valuable suggestion. We have accordingly revised the abstract and added more information, data, and exact experimental values. The revised Abstract was marked by word tracking as follows in line 34 of the manuscript:
“…centers. With AI algorithm-assisted CT screening, the sensitivity of detecting nodules sized 4–5 mm, 6~10 mm, 11~20 mm, and > 20 mm increased by 41%, 11.2%, 10.3%, and 18.7%. Remarkably, the overall sensitivity of detecting varied nodules increased by 20.7% from 67.7% to 88.4%. Furthermore, the sensitivity increased by 18.5% from 72.5% to 91% for detecting ground glass nodules (GGN), which is challenging for radiologists and physicians. The free-response operating characteristic (FROC) AI score was ≥ 0.4, and the AI algorithm standalone CT screening sensitivity reached > 95%...”
Point 2: Ensure all acronyms in the paper are defined
Response: We thank the reviewer’s generous comment and thorough check. We have carefully checked all abbreviations; the full names were noted before they first appeared in the manuscript. The edited abbreviations and full names were marked by word tracking.
Point 3: As part of your background section, provide a breakdown of the concept of imaging, the pathology in question as well as prior AI methods
Response: We appreciate the reviewer’s comment and valuable suggestion. The detection of small lung nodules in chest computed tomography images is an effective method for early detection of lung cancer. With the help of AI algorithms, doctors can more easily identify nodules in hundreds of thin-sliced CT images, significantly improving the efficiency and quality of image readings. We are working to improve the AI algorithm to include the ability to differentiate between benign and malignant nodules, as well as to predict pathology. This would allow the algorithm to provide more accurate and useful information to clinicians, and ultimately, improve patient outcomes. However, our clinicians are still pleased with the ability of current AI to speed up clinical diagnostic and therapeutic procedures and to initiate subsequent treatment plans. It is important to continue to research and develop these capabilities in order to make the most of the potential of AI in the field of medicine.
Point 4: The explicit gap in literature and associated contributions need to be listed
Response: We appreciate the reviewer’s comment and valuable suggestion. We have the ongoing software improvement program including generating example report containing lung-RAD-score, matching images of the same node from different examination dates, differentiating between benign and malignant nodes, detecting lymph nodes, and identifying other lesions that physicians may easily miss. Those issues were presented in the revised Conclusion and was marked by word tracking as follows in line 430 of the manuscript:
“… In order to improve the software in the future, we are working on a function that allows us to match images of the same node from different examination dates, which will enhance our ability to track and compare lung tumors. Our team is working on expanding the capabilities of the V5-MED-LU01 in order to make it more useful for clinicians. The developing functions, including differentiating between benign and malignant nodes, detecting lymph nodes, and identifying other lesions that physicians may easily miss, can improve the accuracy and efficiency of diagnosis and treatment. …”
Point 5: The proposed AI algorithm needs considerable expansion and description, in order to boost understanding and repeatability
Response: We appreciate the reviewer’s comment and valuable suggestion. As we know, V5-MED-LU01 is an AI-based lung nodule detector that uses DICOM image series from CT scanners as input and produces the coordinates and feature attributes of nodules as output. The AI model belongs to the CNN (Convolutional Neural Network) family, and utilizes the architecture of ResNet and UNet. ResNet addresses the vanishing gradient problem by using a reference to the previous layer to compute the output at a given layer. The output from the previous layer, called the residual, is added to the output of the current layer. By configuring different numbers of channels and residual blocks in the module, different ResNet models can be created, such as the deeper 152-layer ResNet-152. Since the input data is a 3D lung image, our ResNet is a 3D version. The U-Net architecture helps to recover feature maps to a higher resolution and has good detection performance. V5-MED-LU01 is a commercially available software package that is protected by patents and intellectual property. As a clinical trial hospital, we are not able to provide further details beyond those provided by the manufacturer. We apologize for any inconvenience this may cause.
Point 6: The resolution on Fig 3 needs to be improved
Response: We thank the reviewer’s generous comment and thorough check. All the figures with proper resolution are packed and sent to editorial office in a separated .zip file. The Fig 3 will as large as 3393 * 4357 pixels.
Point 7: Are there related studies to benchmark your current study with?
Response: We appreciate the reviewer’s comment and valuable suggestion. Currently, there is no large, fully labeled, thin-sliced computed tomography database available for product-to-product comparisons of nodule detection software. As we know, similar products are being validated in clinical trials or are under regulatory approval. Additionally, clinicians may have different interpretations of computed tomography which may have great influence in LROC of AUC. At present, only the AJR product (ClearRead) has published papers for comparison. In their study, 324 thoracic computed tomography scans were collected. The stand-alone detection rate of the AJR product was 89.5%, and the LROC of AUC increased from 0.633 to 0.773. The V5-MED-LU01 used in this study had a stand-alone detection rate of 95.6%, and the LROC of AUC increased from 67.7% to 88.4%. In the subgroup analysis, the detection of small nodules between 4 and 5 mm was more effective by V5-MED-LU01. It would be beneficial to have a publicly available library for objective comparison as a standard verification procedure for nodule detection software. This would allow for more accurate and fair comparisons between different products, and would ultimately help to improve the accuracy and effectiveness of these tools.
Point 8: It may be worth including a flow diagram for your proposed method
Response: We appreciate the reviewer’s comment and valuable suggestion. The manufacturer of V5-MED-LU01 refused to provide a flow diagram of their software. As a clinical trial hospital, we are not able to provide further details beyond those provided by the manufacturer. We apologize for any inconvenience this may cause.
Point 9: the conclusion requires considerable bolstering with explicit points for further work
Response: We appreciate the reviewer’s comment and valuable suggestion. The software V5-MED-LU01 have been implemented in our outpatient clinic. We also have further effort to make it better. We make the feedback opinion from physicians and our further works clearer at the end of discussion. The revised Conclusion was marked by word tracking as follows in line 424 of the manuscript:
“Currently, we use the V5-MED-LU01 service in our outpatient clinic for unreported CT or LDCT studies of the chest, particularly for CT images that patients bring in from other hospitals. It helps to quickly annotate pulmonary nodules and saves time for busy physicians. Physicians are very satisfied with its performance as an immediate aid in the outpatient clinic and as a backup. Additionally, we are conducting a longitudinal study to evaluate changes in lung tumors using the V5-MED-LU01's annotation and measurement functions as the core of the study. In order to improve the software in the future, we are working on a function that allows us to match images of the same node from different examination dates, which will enhance our ability to track and compare lung tumors. Our team is working on expanding the capabilities of the V5-MED-LU01 in order to make it more useful for clinicians. The developing functions, including differentiating between benign and malignant nodes, detecting lymph nodes, and identifying other lesions that physicians may easily miss, can improve the accuracy and efficiency of diagnosis and treatment. These features may be especially useful in busy outpatient clinics and have the potential to have a positive impact on patient care”
Point 10: Refs need considerable bolstering
Response: We appreciate the reviewer’s comment and valuable suggestion. We have thoroughly considered the Conclusion and added more relevant references. The revised Conclusion was marked by word tracking as follows in line 413 of the manuscript:
“…lymph nodes [3]. Besides, the interpretation of AI engagement is also critically affected by the CT image stack input quality, which needs a more sophisticated algorithm to pre-treat the CT images [35]. Since the FP rate can lead to the underestimation of GGN in CT screening, ameliorating the sensitivity and maintaining the specificity of detecting GGN has emerged as a critical medical problem. Nevertheless, data from large scale trails have demonstrated that lung cancer mortality was significantly lessened by the more prevalent LDCT screening [4]; the management of subsequent treatment from a doctor and hospital care should be more seriously considered [36,37]. Collectively, our data fulfill this unmet need.”
Round 2
Reviewer 1 Report
Thank you for considering my suggestions and revising your manuscript accordingly.
Reviewer 3 Report
Thanks for making the changes